# Mycousfurans A and B, Antibacterial Usnic Acid Congeners from the Fungus *Mycosphaerella* sp., Isolated from a Marine Sediment

**DOI:** 10.3390/md17070422

**Published:** 2019-07-19

**Authors:** Jihye Lee, Jusung Lee, Geum Jin Kim, Inho Yang, Weihong Wang, Joo-Won Nam, Hyukjae Choi, Sang-Jip Nam, Heonjoong Kang

**Affiliations:** 1Laboratories of Marine New Drugs, REDONE Seoul, Seoul 08594, Korea; 2Department of Chemistry and Nanoscience, Ewha Womans University, Seoul 03760, Korea; 3Laboratory of Marine Drugs, School of Earth and Environmental Sciences, Seoul National University, NS-80, Seoul 08826, Korea; 4College of Pharmacy, Yeungnam University, Gyeongsan-si, Gyeongsangbukdo 38541, Korea; 5Department of Convergence Study on the Ocean Science and Technology, Korea Maritime and Ocean University, Busan 49112, Korea; 6Research Institute of Oceanography, Seoul National University, NS-80, Seoul 08826, Korea

**Keywords:** usnic acid, mycousfurans, mycousnine, placodiolic acid, *Mycosphaerella* sp., antibacterial activity

## Abstract

Mycousfurans (**1** and **2**), two new usnic acid congeners, along with (−)-mycousnine (**3**), (−)-placodiolic acid (**4**), and (+)-usnic acid (**5**), were isolated using high-performance liquid chromatography-ultraviolet (HPLC-UV)-guided fractionation of extracts of *Mycosphaerella* sp. isolated from a marine sediment. The planar structures of **1** and **2** were elucidated using 1D and 2D NMR spectra. The relative configurations of the stereogenic carbons of **1** and **2** were established via analysis of their nuclear Overhauser spectroscopy (NOESY) spectra, and their absolute configurations were determined using a comparison of experimental and calculated electronic circular dichroism (ECD) spectra. Compounds **1** and **2** were found to have antibacterial activity, showing moderate activity against *Kocuria rhizophila* and *Staphylococcus aureus*.

## 1. Introduction

Dibenzofurans have been isolated from plants, mushrooms, and marine organisms. Although lichens were their first reported natural source, isolation of dibenzofurans from filamentous fungi has been increasingly reported [1]. Usnic acid (UA) is the most representative dibenzofuran natural product and has interesting chemical and pharmacological properties with a broad spectrum of biological activities such as antibacterial, antiviral, anti-inflammatory, antiprotozoal, antifungal, anti-proliferative, phytotoxic, UV filter, and anti-osteoclastogenic activities [2,3,4,5,6]. UA is generally distributed in lichen genera such as *Usnea* (Usneaceae), *Cladonia* (Cladoniaceae), *Hypotrachyna* (Parmeliaceae), *Lecanora* (Lecanoraceae), *Ramalina* (Ramalinaceae), *Evernia*, *Parmelia* (Parmeliaceae), and *Alectoria* (Alectoriaceae) [7]. There are also a few reports on the isolation of UA or its derivatives from non-lichen sources [4].

*Mycosphaerella* is the largest genus of Ascomycota, with more than 10,000 species. *Mycosphaerella* species produce secondary metabolites including rosigenin [8], rubellins A and B [9], (−)-mycousnine, and (+)-isomycousnine [10]. Mycousfuranine and mycousnicdiol have also been recently reported to possess antifungal activity [11]. *Mycosphaerella* species are generally known as foliicolous plant pathogens, isolated from the leaves of plants; however, some species are also found in marine environments. *M. ascopliylli* and *M. pelvetiae* are endophytes of the brown algae *Ascophyllum nodusum* and *Pelvetia canaliculate*, respectively, while *M. apophlaeae* is the symbiont of the rhodophyte, *Apophlaea lyallii* [12,13,14]. In addition, there are recent taxonomy studies demonstrating that *Mycosphaerella* is not just a terrestrial genus but is spread across marine environments as well. For example, *Mycosphaerella* sp. was one of the two dominant fungal communities in samples collected from salt marshes in California Bay and the Atlantic east coast of USA [15,16].

During the course of the chemical analysis of cultured fungal strains, isolated from marine sediments, we isolated two new usnic acid congeners, mycousfurans A and B (**1** and **2**), along with the previously reported compounds, (−)-mycousnine (**3**), (−)-placodiolic acid (**4**), and (+)-usnic acid (**5**), from extracts of *Mycosphaerella* sp. (Figure 1). Herein, we describe the isolation, structural elucidation, and bioactivities of mycousfurans A and B (**1** and **2**).

## 2. Results

### 2.1. Isolation and Structure Elucidation

Compound **1** was obtained as an amorphous yellowish powder, and its molecular formula was determined to be C_18_H_20_O_7_ based on a (+)-high-resolution electrospray ionization mass spectrometry (HRESIMS) *m/z* 349.1302 [M + H]^+^, indicating 9 degrees of unsaturation. The IR spectrum of **1** indicated the presence of hydroxyl (3387, 3232 cm^−1^) and ketone functionalities (1616 cm^−1^), and the UV spectrum showed similar absorption patterns to those of dibenzofuran derivatives. The ^1^H NMR spectrum of **1** showed five methyl singlets (δ_H_ 1.62, 2.04, 2.61, 3.49, 3.81), two doublets at δ_H_ 2.96, *J* = 17.5 Hz (1H) and δ_H_ 3.15, *J* = 17.5 Hz (1H), a singlet of an olefinic proton (δ_H_ 5.55, s), and two singlets of phenolic hydroxyl protons (δ_H_ 9.34, 13.34). The ^13^C NMR, in combination with the heteronuclear single quantum correlation (HSQC) spectrum, showed two ketone carbonyls (δ_C_ 200.5, 201.2), six non-protonated aromatic carbons (δ_C_ 102.0, 106.5, 107.5, 157.1, 159.6, 163.3), five methyl carbons (δ_C_ 7.4, 16.6, 31.3, 50.9, 56.9), two bridgehead quaternary carbons (δ_C_ 57.9, 111.6), a methine sp^2^ carbon (δ_C_ 100.6), and one methylene sp^3^ carbon (δ_C_ 34.3) (Table 1). The heteronuclear multiple bond correlation (HMBC) correlations from the phenolic proton 7-OH (δ_H_ 9.34) to C-6 (δ_C_ 102.0), C-7 (δ_C_ 163.3), and C-8 (δ_C_ 107.5); from the phenolic proton 9-OH (δ_H_ 13.34) to C-8 (δ_C_ 107.5) and C-9a (δ_C_ 106.5); from the methyl protons H_3_-13 (δ_H_ 2.61, s) to C-6 (δ_C_ 102.0) and C-12 (δ_C_ 201.2); and from H_3_-11 (δ_H_ 2.04, s) to C-8 (δ_C_ 107.5), established the substitution pattern of the A ring. The HMBC correlations from H_2_-4 (δ_H_ 2.96, d, *J* = 17.5 and 3.15, d, *J* = 17.5) to C-2 (δ_C_ 106.6) and C-4a (δ_C_ 34.3); from H_3_-15 (δ_H_ 3.81, s) to C-2 (δ_C_ 106.6) and C-3 (δ_C_ 175.4), and from the angular methyl protons H_3_-10 (δ_H_ 1.62, s) to C-1 (δ_C_ 200.5), C-4a (δ_C_ 34.3), and C-9b (δ_C_ 57.9) established the substitution pattern of ring C. Finally, the linkage between C-9a and C-9b was corroborated by the observation of the cross peak from H_3_-10 to C-9a in the HMBC spectrum. Combining the 1D and 2D NMR data and the molecular formula, the presence of an ether linkage between C-4a (δ_C_ 111.6) and C-5a (δ_C_ 157.1) was proposed. Moreover, the HMBC correlation from the methoxyl protons H_3_-14 (δ_H_ 3.49, s) to C-4a permitted the placement of the methoxy group at C-4a, thus completing the structural assignment of **1**, as shown in Figure 2. The NOESY correlations from H_3_-14/H_3_-10 and H_2_-4/H_3_-10 indicated that the B/C ring junction was *cis*-orientated (Appendix A) [10].

Compound **2** was obtained as a yellow amorphous powder. Its molecular formula was determined to be C_18_H_20_O_7_ based on a (+)-HRESIMS *m/z* 349.1305 [M + H]^+^ and ^13^C NMR data. Interpretation of the NMR data revealed that the structure of **2** was almost identical to that of **1**, except that **2** possessed a methyl group at C-6 and an acetyl group at C-8 (Figure 1). The HMBC correlations from H_3_-10 (δ_H_ 1.66, s) to C-9a (δ_C_ 106.1), 9-OH (δ_H_ 9.61, s) to C-8 (δ_C_ 107.1) and C-9a (δ_C_ 106.1), H_3_-13 (δ_H_ 2.73, s) to C-8, and H_3_-11 (δ_H_ 2.02, s) to C-5a (δ_C_ 160.7) supported the positions of the acetyl group at C-8, and consequently placed a methyl group at C-6 (δ_C_ 100.4) (Appendix A). Thus, **2** was defined as a regioisomer of **1**.

The absolute configurations of the stereogenic carbons of **1** were established via the comparison of the experimental electronic circular dichroism (ECD) spectrum with that generated by the computer-assisted ECD calculation. The ECD spectrum of **1** fits well with that of the calculated ECD spectrum of 4a*S* and 9b*R* stereoisomers (Figure 3). Both experimental and calculated ECD spectra of **1** showed the negative absorption in the range from 210 to 235 nm and the positive absorption from 250 to 310 nm (Figure 3). Therefore, the absolute configurations of C-4a and C-9a in **1** was established as 4a*S* and 9b*R*. The experimental ECD spectrum of **2** was almost identical to that of **1**, leading to the conclusion that the absolute configurations of C-4a and C-9a in **2** were 4a*S* and 9b*R*.

Three known UA derivatives were also isolated together with **1** and **2** and they were identified as (−)-mycousnine (**3**) [10], (−)-placodiolic acid (**4**) [17], and (+)-usnic acid (**5**) [18] via comparison of their NMR and MS data with those reported in the literatures.

### 2.2. Bioactivity

Compounds **1**–**5** were tested for their antibacterial activity using three Gram-positive bacteria (*Bacillus substilis* ATCC 6633, *Kocuria rhizophila* ATCC 9341, *Staphylococcus aureus* ATCC 6538) and three Gram-negative bacteria (*Escherichia coli* ATCC 11775, *Salmonella typhimurium* ATCC 14208, *Klebsiella pneumonia* ATCC 4352). Compound **1** exhibited the minimal inhibitory concentration (MIC) values of 8 μg/mL and 32 μg/mL, while **2** exhibited MIC values of 16 μg/mL and 32 μg/mL, against *K. rhizophila* and *S. aureus,* respectively (Table 2). Compounds **1** and **2** showed no antibacterial activity against *B. substilis* and Gram-negative bacteria. Since the MIC values of **3**–**5** indicated stronger antibacterial activity than that of **1** and **2** against Gram-positive bacteria, it was suggested that the substituents in ring C could play a role in the antibacterial activity.

## 3. Materials and Methods

### 3.1. General Experimental Procedures

Optical rotations were measured using an Autopol III (Rudolph Research Analytical, Hackettstown, NJ, USA) polarimeter with a 5-cm cell. ECD spectra were recorded using a Chirascan™-plus CD Spectrometer (Applied Photophysics Ltd., Surrey, UK) and the UV spectra were recorded on a Scinco UVS-2100 spectrophotometer (Sinco, Daejeon, Korea). IR spectra were obtained using a Scimitar 800 FT-IR spectrometer (Varian Inc., Palo Alto, CA, USA). NMR spectra were recorded on a Bruker Avance 700 MHz spectrometer (Bruker Biospin Group, Karlsruhe, Germany); The residual solvent signals of CDCl_3_ (δ_H_ 7.26, δ_C_ 77.0) were referenced for the ^1^H and ^13^C chemical shift values. HRESIMS spectra were obtained using a JEOL JMS-AX505WA mass spectrometer (JEOL Ltd., Tokyo, Japan). Low-resolution LC-MS data were obtained using an Agilent Technologies 6120 quadrupole LC/MS system (Agilent Technologies, Santa Clara, CA, USA) with a reversed-phase C18 column (Phenomenex Luna C18 (2), 50 mm × 4.6 mm, 5 μm) at a flow rate of 1.0 mL/min. Column chromatography separation was performed using a C18 column (40–63 m, ZEO prep 90), eluting with a gradient of methanol and water. The fractions were purified using a WATERS^TM^ (Milford, MA, USA) 1525 binary HPLC (high-performance liquid chromatography) pump, equipped with a WATERS 2489 UV visible detector using a WATERS reversed-phase HPLC Watchers 120 ODS-BP (250 mm × 10 mm, 5 μm) column, eluting with 80% CH_3_CN in H_2_O at flow rate of 2.5 mL/min.

### 3.2. Fungal Material

The strain F8015-2B was isolated from a marine sediment at a 5-m depth in Donghae-si, Gangwon-do, South Korea. The collected sediment was dried on a clean bench for 24 h and then crushed using a sterile spoon. The powder was stamped onto 1/3 marine agar medium and incubated at 27 °C. After two weeks, fungal spores were observed. The spores were cultured via repeated inoculation on potato dextrose agarplates. F8015-2B was identified as *Mycosphaerella* sp. based on a 99.6% (496/498) similarity of 18S rRNA genes to the *Mycosphaerella nawae* strain MY3.

### 3.3. Fermentation, Extraction, and Purifircation

The strain F8015-2B was cultured in 6 × 2.5-L Ultra Yield Flasks (Thomson Instrument Company, Oceanside, CA, USA), each containing 1 L of potato dextrose broth (PDB) dissolved in seawater. The fungus was cultivated on seed agar blocks in 6 × 2.5-L Ultra Yield Flasks, each containing 1 L of PDB dissolved in seawater at 27 °C and 140 rpm in a shaking incubator. After seven days, the mycelia were filtered from the broth using gauze filtration and extracted with acetone and methanol. The broth was extracted with EtOAc and evaporated to obtain the crude extract (4.01 g).

The crude extract was fractionated into eight fractions with a silica gel open column chromatography using a step-gradient with a mixture of CH_2_Cl_2_ and MeOH as an eluent. Fractions 1 (974.9 mg), 2 (280.1 mg), and 3 (420.1 mg) were subjected to a reversed-phase HPLC (Phenomenex luna C18 column, 250 mm × 10 mm, 5 μm, flow rate = 2.0 mL/min) and eluted with 65% CH_3_CN in distilled water to yield **1** (7.5 mg), **2** (3.3 mg), **3** (56.3 mg), **4** (28.7 mg), and **5** (6.8 mg).

*Mycousfuran A* (**1**): amorphous powder, [α]_D_^25^ + 11 (*c* 1.00, CHCl_3_); UV (MeOH) *λ_max_* (log ε) 200 (2.15), 281 (2.09), and 349 (1.24) nm; IR (KBr) ν_max_ 3387, 3232, and 1616 cm^−1^; CD λ_ext_ (MeOH) nm (Δε): 282 (+0.11), 246 (+0.03) [236(0)]; ^1^H and ^13^C NMR data, see Table 1; (+)-HRESIMS, *m/z* 349.1302 [M + H]^+^ (calcd for C_18_H_20_O_7_, 349.1287).

*Mycousfuran B* (**2**): amorphous yellowish powder, [α]_D_^25^ + 15 (*c* 1.00, CHCl_3_); UV (MeOH) *λ_max_* (log ε) 200 (2.15), 281 (2.09), and 349 (1.24) nm; IR (KBr) ν_max_ 3325, 3198, and 1625 cm^−1^; CD λ_ext_ (MeOH) nm (Δε): 282 (+0.12), 246 (+0.06) [236(0)]; ^1^H and ^13^C NMR data, see Table 1; (+)-HRESIMS, *m/z* 349.1305 [M + H]^+^ (calcd for C_18_H_20_O_7_, 349.1287).

### 3.4. Computer-Assisted Conformational Analyses and ECD Calculations

Preliminary conformational analyses of **1** and **2** were performed with Merck Molecular Force Field (MMFF) by Spartan 10 (Wavefunction, Irvine, CA, USA). The two lowest energy conformers of **1** and **2** were geometrically optimized with the B3LYP/6-31G(d,p) level of density functional theory (DFT) in methanol using Gaussian 16 (Expanding the limits of computational chemistry, Wallingford, CT, USA). The computer-assisted ECD calculation was carried out with the B3LYP/6-31G(d,p) level of time-dependent density functional theory (TDDFT). The calculated ECD spectra of **1** and **2** were obtained via visualization of SpecDis version 1.71 (SpecDis, Berlin, Germany) in combination with the calculated ECD spectra of each conformer on the basis of Boltzmann distribution theory and their relative Gibbs free energy.

### 3.5. Antibacterial Activity

Three Gram-positive (*Bacillus substilis* ATCC 6633, *Kocuria rhizophila* ATCC 9341, *Staphylococcus aureus* ATCC 6538) and three Gram-negative (*Escherichia coli* ATCC 11775, *Salmonella typhimurium* ATCC 14208, *Klebsiella pneumonia* ATCC 4352) strains were used. These bacteria were inoculated onto a Mueller–Hinton agar medium and allowed to grow for 24 h at 37 °C. The bacterial colonies were cultivated in 15-mL round-bottom tubes containing 5 mL of Mueller–Hinton broth (MHB) at 37 °C and 220 rpm for 24 h. One hundred microliter aliquots of test compounds and positive controls (vancomycin and ampicillin) at a concentration of 256 µg/mL in DMSO were added to different wells of a 96-well microtiter plate containing 50 µL of MHB. The samples were serially diluted and 50 µL of bacterial MHB medium was adjusted to a concentration of 1/100 dilution. McFarland 0.5% standard was added to the wells. The 96-well microtiter plate was incubated for 24 h at 37 °C. Subsequently, the minimum inhibitory concentration was determined as the concentration of compounds inhibiting bacterial growth [19].

## 4. Conclusions

In conclusion, mycousfurans A and B (**1** and **2**) and other usnic acid congeners, were isolated from a marine sediment-derived fungus *Mycosphaerella* sp. The structures of **1** and **2** were established using 1D and 2D NMR spectra. The absolute configurations of the stereogenic carbons of **1** and **2** were determined using NOESY experiments and a comparison between the experimental and calculated ECD spectra. Compounds **1** and **2** exhibited antibacterial activity against *K. rhizophila*, *S. aureus*, and *E. coli*. The present study is the first report of the antibacterial compounds, produced by *Mycosphaerella* sp., which was isolated from the marine environment.

## Figures and Tables

**Figure 1 marinedrugs-17-00422-f001:**
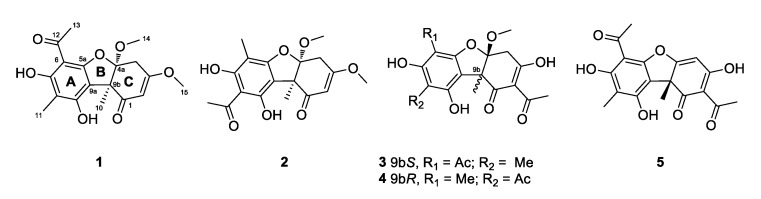
Chemical structures of mycousfurans A and B (**1** and **2**), (−)-mycousnine (**3**), (−)-placodiolic acid (**4**), and (+)-usnic acid (**5**), isolated from *Mycosphaerella* sp.

**Figure 2 marinedrugs-17-00422-f002:**
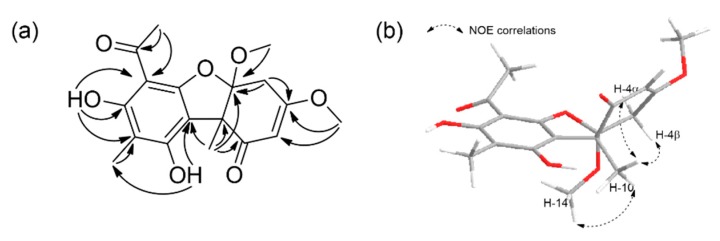
Key HMBC (**a**) and NOESY (**b**) correlations of mycousfuran A (**1**).

**Figure 3 marinedrugs-17-00422-f003:**
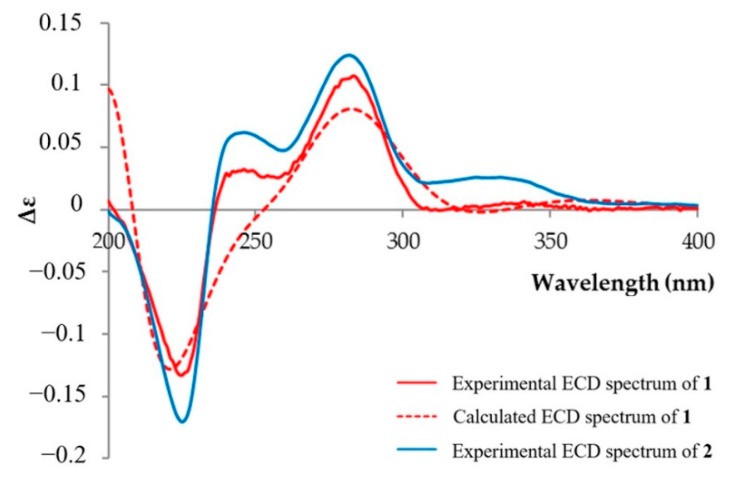
Experimental and calculated ECD spectra of **1** and **2**.

**Table 1 marinedrugs-17-00422-t001:** ^1^H and ^13^C NMR spectroscopic data (700 MHz and 175 MHz in CDCl_3_) for mycousfurans (**1**–**2**).

Position	1		HMBC	2	
	δ_C_, Type	δ_H_, mult. (*J* in Hz)		δ_C_, Type	δ_H_, mult. (*J* in Hz)
1	200.5, C			201.0, C	
2	100.6, CH	5.55, s		100.2, CH	5.55, s
3	175.4, C			175.9, C	
4α 4β	34.3, CH_2_	3.15, d (17.5), 2.96, d (17.5)	2, 4a 2, 4a	34.2, CH_2_	3.20, d (17.5) 2.94, d (17.5)
4a	111.6, C			110.6, C	
5a	157.1, C			160.7, C	
6	102.0, C			100.4, C	
7	163.3, C			165.5, C	
8	107.5, C			107.1, C	
9	159.6, C			156.4, C	
9a	106.5, C			106.1, C	
9b	57.9, C			58.9, C	
10	16.6, CH_3_	1.62, s	1, 4a, 9a, 9b	16.2, CH_3_	1.66, s
11	7.4, CH_3_	2.04, s	8	7.5, CH_3_	2.02, s
12	201.2, C			203.8, C	
13	31.3, CH_3_	2.61, s	6, 12	32.9, CH_3_	2.73, s
14	50.9, CH_3_	3.49, s	4a	50.3, CH_3_	3.47, s
15	56.9, CH_3_	3.81, s	2, 3	56.8, CH_3_	3.84, s
7-OH		13.34, s	6, 7, 8		14.32, s
9-OH		9.34, s	8, 9a		9.61, s

**Table 2 marinedrugs-17-00422-t002:** The MIC values (g/mL)^1^ of **1**–**5** against Gram-positive and Gram-negative bacteria.

Compound	Gram (+) Bacteria			Gram (−) Bacteria		
	*B. subtilis* ATCC 6633	*K. rhizophila* ATCC 9341	*S. aureus* ATCC 6538	*E. coli* ATCC 11775	*S. typhimurium* ATCC 14208	*K. pneumonia* ATCC 4352
**1**	>128	8	32	>128	>128	>128
**2**	>128	16	32	>128	>128	>128
**3**	4	8	4	>128	>128	>128
**4**	4	8	4	>128	>128	>128
**5**	2	8	16	>128	>128	>128
Vancomycin	0.25	0.25	0.5	>128	>128	>128
Ampicillin	0.5	0.25	2	16	8	>128

^1^ Each sample was tested in triplicate and repeated three times.

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
