# Peer review of "Mycousfurans A and B, Antibacterial Usnic Acid Congeners from the Fungus *Mycosphaerella* sp., Isolated from a Marine Sediment"

_marinedrugs, 2019, doi:10.3390/md17070422_

Round 1
Reviewer 1 Report
This paper reports on the isolation and characterization of two new dibenzofurans isolated from a strain of Mycospherella sp. The antimicrobial activity of the new metabolites and three previously described compounds against a panel of gram (+) and (-) bacteria, is also described.
The assignment of the structure of the new compound is based on spectroscopic analysis and mass spectrometry and it is adequate.
The absolute configuration of compounds 1 is deduced from the comparison of the optical rotation value of compound 1 with that of (+)-isomycousnine and (+)-oxymycousnine (Agric. Biol. Chem 1990, 54, 2231-2237). This comparison is not enough to establish the absolute configuration of 1. For example, the only difference between (-)-mycuosnine and (-)-pseudoplacodiolic acid (Tetrahedron let. 1981, 22, 351-352) is the configuration at C-9a.
Minor corrections:
Lines 78 and 98. Reference 15 does not apply in there. I think authors should add reference 10 instead.
Author Response
The absolute configuration of compounds 1 is deduced from the comparison of the optical rotation value of compound 1 with that of (+)-isomycousnine and (+)-oxymycousnine (Agric. Biol. Chem 1990, 54, 2231-2237). This comparison is not enough to establish the absolute configuration of 1. For example, the only difference between (-)-mycuosnine and (-)-pseudoplacodiolic acid (Tetrahedron let. 1981, 22, 351-352) is the configuration at C-9a.
Response) Thank you for your comment. For the more solid evidence for the absolute configurations of 1 and 2, calculated and experimental ECD spectra have been compared as in figure 3.
Minor corrections:
Lines 78 and 98. Reference 15 does not apply in there. I think authors should add reference 10 instead.
Response) Thank you for your comment. It has been corrected.
Reviewer 2 Report
The manuscript can be recommended for publication only after major revision of stereochemistry determination.
Lines 59, 85. Change “HRMS” with “HRESIMS”
Line 79. The determination of absolute configurations can be possible based on optical rotation value just in some cases. Optical rotation of 1 (+11.4) is very different from optical rotation of (+)-oxymycousnine (+106). It means that you cannot suggest that absolute configurations in 1 are identical with (+)-oxymycousnine. Authors have to use another method for verification of absolute configurations of 1 and 2 (for example, comparing of experimental/calculated CD).
Line 169-174. CD data should be added into Experimental section.
Author Response
Lines 59, 85. Change “HRMS” with “HRESIMS”
Response) Thank you for your comment. It has been corrected.
Line 79. The determination of absolute configurations can be possible based on optical rotation value just in some cases. Optical rotation of 1 (+11.4) is very different from optical rotation of (+)-oxymycousnine (+106). It means that you cannot suggest that absolute configurations in 1 are identical with (+)-oxymycousnine. Authors have to use another method for verification of absolute configurations of 1 and 2 (for example, comparing of experimental/calculated CD).
Response) Thank you for your comment. For the more solid evidence for the absolute configurations of 1 and 2, calculated and experimental ECD spectra have been compared as in figure 3.
Line 169-174. CD data should be added into Experimental section.
Response) Thank you for your comment. We have added experimental CD data of 1 and 2 into Experimental section.
Round 2
Reviewer 1 Report
Figure 1 is not complete
Author Response
Thank you for your comment.
File form of Figure 1 was checked in the revised manuscript.
Reviewer 2 Report
The paper was improved compare with original version. Anyway, the main point about stereochemistry of compounds 1 and 2 still confused and disputable. The identical stereochemistry of 1 and 2 is undoubted as you show by CD. Nevertheless, the calculated and measured CD spectra are different. These curves are similar between ~270 and 380 nm, but you cannot ignore huge differences between 230 and 260 nm. Moreover, I cannot find any similarities with literature data [10] for (–)-mycousnine (347 (- 6.0) and 317 (- 8.9) [306 (0)]), that don’t include the most important wave range 230 – 300 nm. So, I’m sure that authors have to give more reliable evidence for stereochemistry of 1 and 2 or remove this information from the paper. Otherwise, this manuscript should be rejected.
Author Response
Thank you for the critical comments on the configurational analysis of compounds 1 and 2. There were mistakes on the description of ECD spectra comparison. As the reviewer clearly pointed out, the original literature of (-)-mycousnine (3) (Sassa and Igarashi, 1990), reference 10 in the revised manuscript, provided ECD absorption values at 347, 317 and 306 nm. Therefore, the direct comparison of measured and calculated ECD spectra of 1 with the records of 3 seemed to be inappropriated. Furthermore, 3 has an additional acetyl unit at C-2 position, and this can make differences in UV absorption and ECD spectra. Particularly, Sassa and Igarashi reported that (+)-oxymycousnine with an acetoxy unit at C-2 rather than acetyl unit at C-2 of 3 showed the opposite sign of specific rotation and different ECD absorption sign compared to those of 3. It seems to be inappropriate to directly compare ECD data of 1 and 3.
Therefore, the description of the comparison of chiroptical properties of 1 and 3 was removed and the speculation of absolute configurations of 1 and 2 based on the comparison with calculated ECD spectra was added.
Round 3
Reviewer 2 Report
This version is satisfactory
Author Response
Thanks for your comments.